Research progress in toxicological effects and mechanism of aflatoxin B1 toxin

Li Congcong congcong_925520@126.com 1
Liu Xiangdong 2
Wu Jiao 1
Ji Xiangbo 3
Xu Qiuliang 1
1 College of Animal Science and Technology, Henan University of Animal Husbandry and Economy , Zhengzhou , Henan , China
2 Huazhong Agricultural University, Key Laboratory of Agricultural Animal Genetics, Breeding and Reproduction of Ministry of Education & Key Lab of Swine Genetics and Breeding of Ministry of Agriculture and Rural Affairs , Wuhan , Hubei , China
3 Henan University of Animal Husbandry and Economy, Henan Key Laboratory of Unconventional Feed Resources Innovative Utilization , Zhengzhou , Henan , China
Abd-Elhakim Yasmina
Electronic publication date: 2022 Aug 4
Publication date: 2022
Volume: 10
Electronic Location ID: e13850
Received 2022 Apr 4; Accepted 2022 Jul 16
Copyright: ©2022 Li et al.
Copyright year: 2022
Copyright holder: Li et al.
License: This is an open access article distributed under the terms of the Creative Commons Attribution License, which permits unrestricted use, distribution, reproduction and adaptation in any medium and for any purpose provided that it is properly attributed. For attribution, the original author(s), title, publication source (PeerJ) and either DOI or URL of the article must be cited.
License URL: https://creativecommons.org/licenses/by/4.0/

Keywords: Mycotoxins, Aflatoxin B1, Toxicities, Mechanisms, Regulation

Funding: Joint Funds of the National Natural Science Foundation of China U2004156 This work was supported by the Joint Funds of the National Natural Science Foundation of China (grant numbers: U2004156). The funders had no role in study design, data collection and analysis, decision to publish, or preparation of the manuscript.

==============================
Fungal contamination of animal feed can severely affect the health of farm animals, and result in considerable economic losses. Certain filamentous fungi or molds produce toxic secondary metabolites known as mycotoxins, of which aflatoxins (AFTs) are considered the most critical dietary risk factor for both humans and animals. AFTs are ubiquitous in the environment, soil, and food crops, and aflatoxin B1(AFB1) has been identified by the World Health Organization (WHO) as one of the most potent natural group 1A carcinogen. We reviewed the literature on the toxic effects of AFB1 in humans and animals along with its toxicokinetic properties. The damage induced by AFB1 in cells and tissues is mainly achieved through cell cycle arrest and inhibition of cell proliferation, and the induction of apoptosis, oxidative stress, endoplasmic reticulum (ER) stress and autophagy. In addition, numerous coding genes and non-coding RNAs have been identified that regulate AFB1 toxicity. This review is a summary of the current research on the complexity of AFB1 toxicity, and provides insights into the molecular mechanisms as well as the phenotypic characteristics.

Introduction

Due to global climate change, mycotoxin-producing fungal strains that were endemic in the tropical-subtropical climate zones have also appeared in the temperate zones. Mycotoxins are secondary metabolites produced by filamentous fungi or mold present in the soil, grain, forage, and silage. They are non-essential for fungal growth and reproduction but are capable of inducing biochemical, physiological and pathological changes in many species (Wang et al., 2018a). The Food and Agricultural Organization (FAO) estimates that around 25% of the global agricultural productions and derived food products are contaminated with mycotoxins, which render an estimated 50 million tons of food inedible, resulting in severe economic losses each year (Iheshiulor et al., 2011). However, the current detectable rate of mycotoxins is as high as 60–80%, which is considerably higher than the FAO estimate of 25% (Eskola et al., 2020). Aflatoxin (AFT) is mainly produced by Aspergillus flavus and A. parasiticus, and is a derivative of difuranoxano-naphtho-ketone consisting of a difuran ring and a coumarin ring. AFTs are produced at temperatures between 10–40 °C, pH 3.0–8.5, moisture content 18–20%, and water activity > 0.82 (Iqbal, Asi & Arino, 2013). Due to their stable structure, AFTs are heat-resistant and can only be degraded above 280 °C (Sun, 2017). In addition, AFTs are insoluble in water but soluble in methanol and other organic solvents. AFTs can contaminate agricultural produce during growth, harvest, transportation, and storage. Current food processing practices and conventional storage conditions cannot completely eliminate AFT contamination from the food supply chain. The products with longer storage, such as corn, soybean, wheat, peanuts, nuts, and dried fruits, are particularly susceptible to AFT contamination (Battilani et al., 2016). In addition, poor post-harvest management can lead to rapid deterioration of the nutritional quality or digestibility of the agricultural produce due to mold contamination, which causes discoloration and degradation of lipids and proteins. Molds also produce volatile metabolites that result in unpleasant odors, thereby affecting grain intake by livestock and poultry, eventually leading to feed refusal and emesis (Magan & Aldred, 2007).

According to their fluorescence, the AFTs are broadly classified into AFB1 and AFB2 (blue), and AFG1 and AFG2 (green) types (Dalvi, 1986). AFM1, AFM2, AFB2a, AFG2a, aflatoxicol (AFL), AFP1, AFQ1 and AFH1 are derivatives of AFTs. AFB1 is considered to be the most toxic AFT (Wong & Hsieh, 1976; Bondy & Pestka, 2000; Hernandez-Mendoza, Garcia & Steele, 2009), with 10-fold and 68-fold higher toxicity compared to that of potassium cyanide and arsenic respectively (Li, 2002), and is primarily responsible for food and feed contamination (Hernandez-Mendoza, Garcia & Steele, 2009). AFB1 has also been identified by the WHO and the International Agency for Research on Cancer, IARC (1993) as one of the most potent natural group 1A carcinogens (Heinrich, 2003). It is ubiquitous in the environment, soil, animal feed and food crops, and is highly toxic to human beings, livestock and poultry. Following ingestion of contaminated feed, AFB1 promotes inflammation and necrosis of liver cells and intestinal cells, impairs liver function (Monson et al., 2014; Zhang et al., 2016), decreases lymphocyte activity and increases thymocyte apoptosis. These changes eventually reduce livestock productivity (Theumer et al., 2003; Peng et al., 2016), and result in huge economic losses. Furthermore, AFB1 can easily enter the human food chain through contaminated meat, eggs, and milk (Kumar et al., 2017). Long-term exposure to sublethal doses of AFT can weaken immunity and lead to nutritional disorders, whereas the mortality rate due to acute exposure at high doses is close to 25% (Liu & Wu, 2010; Verheecke, Liboz & Mathieu, 2016). Death due to acute or chronic AFB1 poisoning has been reported in India, Kenya, and other developing countries (Krishnamachari et al., 1975; Probst, Njapau & Cotty, 2007). Studies also show that AFB1 can induce carcinogenesis in the liver, stomach, lung, kidney, rectal colon, breast, and gallbladder (Harrison, Carvajal & Garner, 1993; Wang & Groopman, 1999; Eom et al., 2013; Cui et al., 2015; Yip et al., 2017; Costa, Lima & Santos, 2021), and 4.6–28.2% of hepatocellular carcinoma (HCC) cases are the result of AFT exposure (Kucukcakan & Hayrulai-Musliu, 2015).

Despite reports of liver toxicity and immunotoxicity in humans and animals due to AFB1 poisoning, there is no systematic review on the hepatotoxicity, enterotoxicity, nephrotoxicity, immunotoxicity, neurotoxicity and reproductive toxicity of AFB1. In this review, we have discussed the mechanisms under the toxic effects of AFB1 in order to provide a reliable reference for further research in animal husbandry, mycotoxins, and the treatment of toxin-related diseases in humans, livestock and poultry.

Survey Methodology

PubMed and Web of Science databases were searched for relevant articles. A total of 7,715 articles appeared Pubmed database using “aflatoxin B1” as the search term and the date of publication from 1975/1/1 to 2022/3/1 . After narrowing the search with keywords including “toxicokinetics of aflatoxin B1”, “hepatotoxicity of aflatoxin B1”, “enterotoxicity of aflatoxin B1”, nephrotoxicity of aflatoxin B1, “neurotoxicity of aflatoxin B1”, “immunotoxicity of aflatoxin B1”, “reproductive toxicity of aflatoxin B1”, “aflatoxin B1 and COX2”, “aflatoxin B1 and Nrf2”, “aflatoxin B1 and p53”, “aflatoxin B1 and microRNA”, “aflatoxin B1 and lncRNA”, and “aflatoxin B1 and livestock”, 1,733 studies were obtained. With “aflatoxin B1” as the search term, 11,741 articles published between 1975–2022 appeared in the Web of Science database, of which 1,514 articles were selected as above. After removing duplicate articles and the articles with little relevance, 137 articles were selected for the review.

Toxicokinetics of AFB1

AFB1 exposure occurs through dietary intake, skin contact and inhalation of contaminated dust. More than 80% of the ingested AFB1 is absorbed in the duodenum and the jejunum through passive transport (Grenier & Applegate, 2013), and accumulates thereafter in the liver, kidney and spleen, although the main target organ of AFB1 is undoubtedly the liver. Toxic effects of AFB1 have been observed in the liver, gastrointestinal tract, nervous system, immune cells, and reproductive organs.

Figure 1 Overview of AFB1 toxicokinetics.

AFB1 is metabolized in humans and animals by cytochrome P450 enzymes, and its metabolites include AFM1, AFP1, AFQ1 and AFBO. AFM1 accumulates in milk, whereas other metabolites are excreted through urine, feces, bile and can also enter the bloodstream. AFB1, aflatoxin B1; AFM1, aflatoxin M1; AFP1, aflatoxin P1; AFQ1, aflatoxin Q1; AFBO, aflatoxin B1-8,9-epoxide; GST, glutathione-S-transferase; AFB1-FAPy, AFB1-formamidopyridine adduct.

AFB1 is metabolized in the liver by P450 enzymes into the carcinogen AFB1-8,9-epoxide (AFBO), which includes the endo-8,9-epoxide (endo-AFBO) and exo-8,9-epoxide (exo-AFBO) isomers (Fig. 1). CYP1A2 and CYP3A4 enzymes play critical roles in the activation of AFB1 in human liver (Gallagher et al., 1994; Gallagher et al., 1996), and bovine hepatocytes (Kuilman, Maas & Fink-Gremmels, 2000). CYP1A2 oxidizes AFB1 into endo- and exo- AFBO and AFM1, whereas CYP3A4 catalyzes AFB1 oxidation into exo- AFBO and AFQ1. CYP2A13, CYP2A3 and CYP321A1 decompose AFB1 into AFP1 (Rushing & Selim, 2019). Metabolism of AFB1 by the supernatant fraction of liver homogenates from different species has shown that AFQ1 is the major metabolite produced in human, monkey and rat liver, and AFP1 is major metabolite produced in the human, monkey and mouse liver. On the other hand, duck liver homogenate metabolized AFB1 to AFL (Roebuck & Wogan, 1977). AFM1 and AFB1-dihydrodiol are the major metabolites in bovine hepatocytes (Kuilman, Maas & Fink-Gremmels, 2000). AFM1 was originally isolated and identified as an AFB1 metabolite in milk (Goto et al., 1986). AFP1 produces an oxidative metabolite of dihydroxyaflatoxin, which is excreted directly or as a glucuronic acid conjugate in bile (Eaton et al., 1988; Holeski et al., 1987). Furthermore, AFP1, AFM1, AFQ1 and AFL form glucuronide and sulfate conjugates (Coppock, Christian & Jacobsen, 2018), and AFP1, AFM1, AFQ1 and AFB1-N7-guanine have been detected in humans following AFB1 exposure (Groopman et al., 1985; Ross et al., 1992; Mykkanen et al., 2005). The different AFB1 metabolites are mainly expelled via feces and urine (Dohnal, Wu & Kuca, 2014). AFBO binds covalently to N7 on guanine to form AFB1-N7-guanine adducts in the DNA double helix (Iyer et al., 1994), resulting in point mutations that may drive carcinogenesis (Lin et al., 2014). The common point mutation caused by the AFB1-N7-guanine adduct is a G →T transversion (Schermerhorn & Delaney, 2014; Lin et al., 2016). Since exo-AFBO has a significantly higher affinity for guanine residues compared to endo-AFBO, it is considered to be the major carcinogenic metabolite. The AFB1-N7-guanine adduct forms an open ring structure under mild alkaline conditions, resulting in a stable AFB1-formamidopyridine adduct (AFB1-FAPy) (Smela et al., 2001) that is excreted with urine. Both isomers of AFBO are detoxified through glutathione (GSH) conjugation by glutathione-S-transferase. In addition, AFBO can be converted to AFT-mercapturic acid by GST, or to AFT-glucosiduronic acid by AFT-dihydropyridine, followed by the formation of GSH conjugates (Bryden, 2012). However, AFBO can also form adducts with serum albumin by covalently binding to the ɛ-amino group of lysine, which remains in circulation. Given its highly unstable nature, AFBO can spontaneously hydrolyze into AFB1 dihydrodiol, which can cause tissue damage, inflammation, and excessive cell proliferation by conjugating with different proteins, eventually promoting carcinogenesis. One of the most common AFB1-induced mutations in human hepatocytes is a G → T transversion in codon 249 of the p53 gene, which causes a 249Arg → 249Ser mutation in the encoded protein (Foster, Eisenstadt & Miller, 1983; Hollstein et al., 1991; Soini et al., 1996). In addition, AFB1 and AFBO can epigenetically increase the mutation rate of the p53 gene by methylating the CpG site in codon 248 (Narkwa, Blackbourn & Mutocheluh, 2017). The p53 gene is a tumor suppressor that is frequently mutated in human cancers, and the mutations promote tumor development by inhibiting apoptosis and increasing proliferation rates. AFBO can also induce mutations indirectly by binding to and damaging DNA repair enzymes (Weng et al., 2017). Furthermore, AFB1 metabolism by the P450 enzyme also generates reactive oxygen species (ROS), such as hydroxyl free radicals, hydrogen peroxide and other free radicals that damage cell membrane and macromolecules (Kucukcakan & Hayrulai-Musliu, 2015). The complex metabolic process of AFB1 is a major determinant of its potent toxicity. Since the formation of DNA adducts by AFB1, and its metabolites can activate proto-oncogenes, quantitative analysis of AFB1-DNA adducts is an important indicator of AFB1 toxicity.

Effect of AFB1 on Livestock and Poultry

The susceptibility of livestock to AFB1 differs across species. Monogastric animals are more susceptible to AFB1 compared to the ruminants since the gut microbes in the latter can metabolize mycotoxins. Long-term exposure to low levels of AFB1 is more common in livestock as opposed to acute poisoning (Fig. 2). Pigs are especially susceptible AFB1 poisoning, and long-term exposure to low levels of dietary AFB1 can inhibit their growth, impair digestive function, and disrupt the intestinal barrier by decreasing SOD activity, and increasing production of pro-inflammatory cytokines such as TNFα, IL1 and TGFβ (Pu et al., 2021). Nevertheless, there are reports of AFB1 poisoning in ruminants as well. Cattle fed with AFB1-spiked fodder showed behavioral changes such as depression and anorexia. AFB1 significantly increased the serum levels of alanine aminotransferase (ALT), aspartate aminotransferase (AST), alkaline phosphatase (ALP), serum creatinine (SCR), catalase (CAT) and malondialdehyde (MDA) in the affected cattle, and decreased that of total protein (TP), magnesium (Mg) and glutathione (GSH). Furthermore, autopsy of the poisoned cattle showed hepatomegaly, gallbladder enlargement, and intestinal and renal hyperemia (Elgioushy et al., 2020). AFB1 contamination of poultry feed can reduce reproductive capacity, hatching rate, chick weight, growth rate, production rate and quality of meat and eggs, and increase susceptibility to diseases and mortality rate (Pandey & Chauhan, 2007; Fouad et al., 2019). In addition, the younger broilers were more susceptible to AFB1 compared to older animals, which could be due to deficient detoxification mechanisms in the former (Wang et al., 2018b). In conclusion, exposure to AFB1can significantly affect the quality and productivity of livestock and poultry by altering their physiological and biochemical indices. Chronic exposure in particular leads to the accumulation of AFB1, and consumption of these contaminated products can adversely affect human health.

Figure 2 Harmful effects of AFB1 on livestock and poultry.

Pigs, ducks, chickens and cattle are particularly susceptible to AFB1, which mainly affects the small intestine, liver, thymus, kidney, gallbladder, etc. Made in ©BioRender (https://biorender.com/).

AFB1-Induced Toxicity

AFB1 toxicity depends the dose, exposure duration, administration mode, solvent, species, gender and target organs, and is summarized in Tables 1 and 2.

Table 1 The toxicities of AFB1 have been reported in vivo.

The hepatotoxicity, enterotoxicity, nephrotoxicity, immunotoxicity, neurotoxicity and reproductive toxicity of AFB1 have been reported in vivo.

Type of toxicity	Doses	Time	Species	Gender	Route	Solvent	Signaling pathway/Cytokines/ potential target molecule	References	
Hepatoxicity	1 mg/kg (b.w.)	Every other day for 4 weeks	mice	–	Gavage (i.g.)	corn oil	COX2, p10 and IL1β↑ (protein)	Zhang et al. (2019a)	
	1 mg/kg (b.w.)	Daily for 4, 6, 10 weeks	rats	male	Intraperitoneal (i.p.)	DMSO	TNFα, IL1α and PCNA ↑ (protein)	Singh, Maurya & Trigun (2015)	
	0.375, 0.75 and 1.5 mg/kg (b.w.)	Daily for 30 days	mice	male	Gavage (i.g.)	olive oil	Cyt-c, Bax, p53 and caspase-3/9 ↑ (protein & mRNA)	Xu et al. (2021)	
	250 µg/kg (b.w.)	5 days per week over 4 weeks or 8 weeks	rats	female	Gavage (i.g.)	olive oil	–	Ali et al. (2021)	
	0.25, 0.75, and 1.5 mg/kg (b.w.)	Daily for 7 days	rats	male	Gavage (i.g.)	corn oil	p53 signaling pathway	Lu et al. (2013)	
	150 µg/kg feed	Daily for 14 days	rats	male	Feed	methanol	–	Qian et al. (2016)	
	0.32 mg/kg (b.w.)	Daily for 12 days	rats	male	Feed	–	–	Zhang et al. (2011)	
	0.25, 0.5 or 1.0 mg/kg (b.w.)	Daily for 7 days	rats	male	Gavage (i.g.)	olive oil	Ahr, Lipc and Lcat ↓ (protein & mRNA) , Scarb1 ↑ (protein & mRNA)	Rotimi et al. (2017)	
	1 mg/kg (b.w.)	Daily for 1, 4 or 7 days	rats	male	Gavage (i.g.)	olive oil	Nrf2, Hmgcoa, and Acc ↓ (mRNA) (day 1)	Rotimi et al. (2019)	
Enterotoxicity	0.6 mg/kg feed	Daily for 21 days	broilers	–	Feed	methanol	FAS, FASL, TNFα, TNF-R1, GRP78/94, and caspase-3/8/10 ↑ (mRNA)	Zheng et al. (2017)	
	0.3 mg/kg (b.w.)	Daily for 28 days	mice	male	Gavage (i.g.)	methanol	tight junction proteins (claudin-1, zonula occludens-1) ↓ (immunohistochemical staining)	Gao et al. (2021)	
	5, 25, 75 µg /kg (b.w.)	5 days per week for 4 weeks	rats	male	Gavage (i.g.)	DMSO	–	Wang et al. (2016a)	
	5, 25, 75 µg /kg (b.w.)	5 days per week for 5 weeks	rats	male	Gavage (i.g.)	DMSO	–	Zhou, Tang & Wang (2021)	
	5 mg/kg feed	Daily for 42 days	broilers	male	Feed	methanol	CYP1A1, CYP1A2, CYP2A6 and CYP3A4 ↑, GSTA3, GSTA4 and EPHX1 ↓ (mRNA)	Wang et al. (2018b)	
	22.02 ppb	–	broilers	male	Feed	–	–	Kana, Teguia & Tchoumboue (2010)	
	2.91 to 120.02 ug/kg feed	Daily for 0-14 days	ducks	male & female	Feed	Contaminated maize	–	Feng et al. (2017)	
	2.03 to 153.12 ug/kg feed	Daily for 15–35 days	ducks	male & female	Feed	Contaminated maize	–	Feng et al. (2017)	
	0.07, 0.75 mg/kg feed	Twice a day for 4 weeks	broilers	male	Feed	basal diet	–	Yunus et al. (2011)	
	1 ppm	Daily for 28 days	broilers	–	Feed	diet	–	Kumar & Balachandran (2009)	
	0.6 mg/kg feed	Daily for 21 days	broilers	–	Feed	methanol	TLR2-2, TLR-4, and TLR7 ↓ (mRNA)	Wang et al. (2018c)	
	0.6 mg/kg feed	Daily for 21 days	broilers	–	Feed	methanol	ATM, p53, Chk2 and p21 ↑, Mdm2, cdc25C, cdc2, cyclin B, MDM2 and PCNA ↓ (protein & mRNA)	Yin et al. (2016)	
Nephrotoxicity	300 µg/kg (b.w.)	Daily for 30 days	mice	male	Gavage (i.g.)	DMSO	–	Huang et al. (2019)	
	5, 7.5, 10 µg /kg (b.w.)	8/16/24 h	carp	–	Gavage (i.g.)	feed	Keap1-Nrf2 pathway ↑ (mRNA)	Kövesi et al. (2020)	
	0.5 g/kg (b.w.)	Daily for 28 days	mice	male	Gavage (i.g.)	1%DMSO	L-proline ↓ proline dehydrogenase (PRODH) and Bax, Caspase-3 ↑, Bcl-2 ↓ (protein & mRNA)	Li et al. (2018)	
	40 ppm	Daily for 8 weeks	rats	male	Feed	DMSO	GPx and GSH ↓, PER ↑ (activity)	Rotimi et al. (2018)	
Neurotoxicity	300 µg/kg (b.w.)	Daily for 30 days	mice	male	Gavage (i.g.)	DMSO	–	Huang et al. (2020)	
	0.025 mg/kg (b.w.)	Daily for 30 days, 60 days and 90 days	rats	male	Gavage (i.g.)	olive oil	CAT and SOD ↓, ACP, ALP, AST and LDH ↑ (activity)	Alsayyah et al. (2019)	
	0, 0.1, 0.5, 1, 2, 5, and 10 µM	24 /48 h	24/12 hpf zebrafish embryos	–	–	DMSO	gfap, mbp, and olig2 ↓ (protein), caspase 3a/8/9 and p53 ↑ (mRNA)	Park et al. (2020)	
	0, 0.25, 0.5, 1.0, and 2.0 µM	24 h	24, 28, 72, and 96 hpf zebrafish embryos	–	–	DMSO	Gly , Glu, and GABA ↑ (Metabolic Profiles)	Zuberi et al. (2019)	
	15.75 µg/kg (b.w.)	once weekly for 8 weeks	rats	female	Gavage (i.g.)	olive oil	–	Bahey, Abd Elaziz & Gadalla (2015)	
Immunotoxicity	0.25 mg/kg (b.w.)	Daily for 15 days	mice	female	Gavage (i.g.)	ethanol	IFN-γ and TNFα↓, IL4 ↑ (mRNA)	Jebali et al. (2015)	
	1 mg/kg (b.w.)	Daily for 4/6/10 weeks	rats	male	Intraperitoneal (i.p.)	DMSO	TNF-α, IL-1α and PCNA ↑ (mRNA)	Qin et al. (2016)	
	0.15, 0.3, and 0.6 mg /kg diet	Daily for 21 days	broilers	–	Feed	methanol	GSH-Px, GR, and GSH ↓, MDA ↑ (activity)	Chen et al. (2013)	
	385, 867 or 1807 µg/kg feed	Daily for 28 days	pigs	male	Feed	–	TNF-α, IL-1β, IL-6, IFN-γ and IL10 ↑ (mRNA)	Meissonnier et al. (2008)	
	0.6 mg/kg feed	Daily for 21 days	broilers	–	Feed	methanol	Bax, Bak, and cytC ↑, Bcl-2 and Bcl-xL ↓, FasL, Fas and FADD ↑ (mRNA)	Peng et al. (2016)	
	0.6 mg/kg feed	Daily for 21 days	broilers	–	Feed	methanol	Bax, Bak, cytC, caspase-9, Apaf-1, and caspase-3 ↑, Bcl-2 and Bcl-xL ↓, Grp78/Bip, Grp94 and CaM ↑ (mRNA)	Yuan et al. (2016)	
	0.6 mg/kg feed	Daily for 21 days	broilers	–	Feed	methanol	ATM-Chk2-cdc25-cyclin B/cdc2 pathway, ATM-Chk2-cdc25-cyclin D/CDK6 pathway and ATM-Chk2-p21-cyclin D/CDK6 pathway	Hu et al. (2018)	
	100 ppm	120 min	rats	male	Intraperitoneal (i.p.)	–	TNF-α↑ (protein)	Mohammad et al. (2017)	
	5 mg/kg	24 h	chicks	–	Gavage (i.g.)	–	MDA ↑, SOD and GSH-PX ↓ (activity), NLRP3, COX-2, iNOS, IL-6, IL-1β, TNF-α, caspase-1, caspase-3, and caspase-11 ↑ (mRNA)	Gao et al. (2022)	
Reproductive toxicity	20 µg/kg (b.w.)	Daily for 7/14/21 days	mice	male	Intraperitoneal (i.p.)	corn oil and ethanol (95:5, v/v)	Bcl-2 ↓, Bax, p53, and caspase-3 ↑ (mRNA & protein)	Yasin, Mazdak & Mino (2018)	
	0.375, 0.75, or 1.5 mg/kg (b.w.)	Daily for 30 days	mice	male	Gavage (i.g.)	corn oil	p38 MAPK signaling pathway, Occludin, N-cadherin, and Connexin 43 ↓, cyt-c and caspase-3 ↑ (protein)	Huang et al. (2021)	
	6 mg/kg	6 h	mice	female	Intraperitoneal (i.p.)	DMSO	CYP1A2 and CYP3A4 ↑, GSTA1/2 ↓ (protein)	Sriwattanapong et al. (2017)	
	0.5 and 5 mg/kg feed	Daily for 8 weeks (only for mother)	rats	female	Feed	–	–	Rotimi et al. (2021)	
	1.0 mg/kg(b.w.)	24 h	rams	male	Gavage (i.g.)	4% ethanol	StAR, 3β-HSD, CYP11A1, and CYP17A1 ↑ (mRNA), Caspase3 ↓, Bax ↑ (mRNA)	Lin et al. (2022)	
	100 ppb, 200 ppb, and 400 ppb	Daily for 10 weeks	cockerels	male	Feed	–	IgM and IgG ↓ (antibody titers), LH, prolactin, and testosterone ↓	Ashraf et al. (2022)	

Table 2 The toxicities of AFB1 have been reported in vitro.

The hepatotoxicity, enterotoxicity, nephrotoxicity, immunotoxicity, neurotoxicity and reproductive toxicity of AFB1 have been reported in vitro.

Type of toxicity	Doses	Time	Cell type	Solvent	Signaling pathway/Cytokines/potential target molecule	References	
Hepatoxicity	0.05, 0.25 µM	24 h	human liver HepaRG cells	corn oil	caspase 1/3, COX2, and IL-1β↑ (mRNA)	Zhang et al. (2019a)	
	1 µM	24 h	human liver HepaRG cells	corn oil	COX2 and IL1β↑ (protein)	Zhang et al. (2019a)	
	0.5, 1, 2.5 and 5 µmol/L	6 /12 h	primary broiler hepatocytes	DMSO	Nrf2 ↑ (mRNA), caspase-3/9 ↑ (protein)	Liu & Wang (2016)	
Enterotoxicity	0.5 and 4 µg/mL	48 h	Caco-2	DMSO	tight junction proteins (claudin-1/3/4, zonula occludens-1)↓ (mRNA & protein)	Gao et al. (2021)	
Nephrotoxicity	1, 2 and 6 µg/mL	12 /24 /48 h	HEK-293T	DMSO	p21 ↑, PLK1, MYC, and PLD1 ↓ (mRNA & protein)	Huang et al. (2019)	
	5, 10, 50, 100 mg/L	48 h	HEK-293T	DMSO	PRODH and Bax, Caspase-3 ↑, Bcl-2 ↓ (protein & mRNA)	Li et al. (2018)	
	16.3, 32.60, 48.9 µM	24 h	HEK-293T	complete culture medium	Nrf-2, SOD2, GPx, and Hsp70 ↑ (protein), OGG1 ↑ (mRNA), p53,NF-κb, c-IAP and Bax ↑ (protein), caspase 9/3/7↑ and caspase 8 ↓ (activity)	Dlamini et al. (2021)	
Neurotoxicity	1, 5, 10, 20 µg/mL	24 /48 h	IMR-32	DMSO	–	Huang et al. (2020)	
	2 µg/mL and 6 µg/mL	24 h	IMR-32	DMSO	NOX2 ↑, OXR1, SOD1 and SOD2 ↓, PARP1, BRCA2, and RAD51 ↓, CDKN1A, CDKN2C, and CDKN2D ↑, CASP3 and BAX ↑ (mRNA)	Huang et al. (2020)	
	1, 2, 5, 10, 20, or 50 µM; 0.3, 0.6, 1.5, 3.1, 6.2, or 15 µg/mL	36 /48 h	NHA-SV40LT	DMSO	AKT and ERK1/2 signaling pathways, BAX, BAK, and cytochrome c ↑ (protein)	Park et al. (2020)	
Immunotoxicity	20 ng/mL	1/3/6/12/ 24/48 h	CHME5	96% ethanol	TLRs, MyD88, NFκB, and CxCr4 ↑ (mRNA), caspase-3/7 ↑ (activity), IFN-γ and GM-CSF ↑ (protein)	Mehrzad, Hosseinkhani & Malvandi (2018a)	
	0.02, 0.04, 0.08 and 0.16 µg/mL	48 h	3D4/21	DMSO	JAK2/STAT3 pathway, IL6 and TNFα↑ (mRNA), DNMT1/3a ↑ (mRNA & protein)	Zhou et al. (2019)	
	10 ng/mL	2/12 h	MDDCs	96% ethanol	cytochrome P450 (CYP) family, MyD88, NF-KB, TNF-α, TLR2, TLR4, COX-2, HLA-DR, CCR7, CD209, LFA3 and CD16 ↑,AhR, TGF-β, CD11c and CD64 ↓ (mRNA)	Mehrzad et al. (2018b)	
	0.01, 0.02, 0.04, 0.08, 0.16, 0.32, and 0.64 mg/mL	24 /48 h	3D4/21	DMSO	NF-kB signaling pathway, IL6 and TNFα↑ (mRNA)	Hou et al. (2018)	
	0.16, 16, 1600, 160000 nmol/L	24 /48 h	swine alveolar macrophages (SAM)	DMSO	–	Pang, Chiang & Chang (2020)	
	3.125, 6.25, 12.5, 25, 50 and 100 µm	24 /48 h	RAW264.7	DMSO	NOS2, TNF-α and CXCL2 ↑, CD86 ↓ (mRNA)	Ma et al. (2021)	
	0.4, 0.8, 1.6, and 3.2 µg/mL	90 min	chicken heterophils	–	NADPH oxidase and p38 signaling pathways, glycolysis pathway	Gao et al. (2022)	
Reproductive toxicity	0.1, 1, 10 and 100 µM	2 /4 h	bull spermatozoa	DMSO	–	Komsky-Elbaz, Saktsier & Roth (2018)	
	10 and 50 µM	27 /44 h	porcine oocyte	DMSO	H3K27me3 and H3K4me2 ↓, H3K9me3 ↑ (fluorescence intensity), LC3 ↑ (protein), ATG3, ATG5 and ATG7 ↑ (mRNA), Bak, Bax, and Bcl-xl ↑ (mRNA)	Liu et al. (2015a)	
	0.01, 0.1, 1nM	7 days	porcine embryos	DMSO	Bax and Casp3 ↑, Bcl2 and Bcl-xl ↓, Lc3 and Beclin1 ↑ (mRNA)	Shin et al. (2018)	
	0.01, 0.1, 1, 10 and 100 nM	24 h	JEG-3	DMSO	PKC-ERK signaling axis, COX2 ↑ (protein)	Zhu, Tan & Leung (2016)	

Hepatotoxicity

Liver is the main target organ for AFB1. Dietary supplementation of AFB1 in rats lead to irreversible liver damage in a dose-dependent manner (Qian et al., 2016; Lu et al., 2013) by inducing fat deposition, fatty acid oxidation (Zhang et al., 2011) and telomere shortening (Ali et al., 2021). Recent studies have also shown that AFB1 can trigger massive production of reactive oxygen species (ROS) in the liver cells, leading to oxidative stress, inflammation, and liver damage (Singh, Maurya & Trigun, 2015). Furthermore, AFB1 exposure enhanced apoptosis of liver cells, activated the resident Kupffer cells, and promoted an inflammatory response in the liver through dephosphorylated-cyclooxygenase-2 (COX2) (Zhang et al., 2019a). Prolonged exposure to AFB1 disrupted lipid and lipoprotein metabolism (Rotimi et al., 2017), and resulted in extensive damage to mitochondrial lipids and reduced antioxidant capacity in the rat liver (Rotimi et al., 2019). Likewise, mice exposed to AFB1 showed mitochondrial dysfunction and increased rates of mitochondria-dependent apoptosis in the liver (Xu et al., 2021). In one study, primary broiler hepatocytes (PBHs) treated with different concentrations of AFB1 showed mitochondrial dysfunction, oxidative stress, and ROS-dependent mitochondrial apoptosis through the nuclear factor-erythroid 2-related factor-2 (Nrf2) signaling pathway (Liu & Wang, 2016). Epidemiological studies have shown that AFB1 is one of the important risk factors of primary liver cancer (Kucukcakan & Hayrulai-Musliu, 2015). Furthermore, there is evidence that AFB1 and chronic hepatitis B virus can synergistically induce mutations in the p53 gene and initiate liver cancer (Liu et al., 2012). To summarize, AFB1 can induce liver damage and even liver cancer by inducing oxidative stress, inflammation, and mitochondrial dysfunction by targeting the p53, ROS, COX2, Nrf2 and other signaling pathways.

Enterotoxicity

The gut barrier function maintains the homeostasis between the resident immune cells and commensal microorganisms via the intestinal epithelial cells (IECs) (Peterson & Artis, 2014). Long-term exposure to AFB1 has been associated with chronic intestinal diseases. AFB1 caused intestinal mucosal injury and inhibited IECs proliferation in mice (Gaikwad & Pillai, 2004). In addition, AFB1 and AFM1 can damage the intestinal barrier in mice through clathrin-mediated endocytosis through synergistic and additive interactions (Gao et al., 2021). AFB1 also altered the composition of the intestinal microbiota of male F344 rats in a dose-dependent manner, and significantly decreased the abundance of the probiotic lactic acid bacteria (Wang et al., 2016a). The microbial-related metabolic changes in the gut microbiota of these AFB1-treated rats were analyzed by ion fragmentation spectroscopy. AFB1 significantly increased the number of inflammatory fecal liposomes, and altered intestinal microbiota-dependent biliary cholesterol metabolism, degradation of bilirubin and fatty acids, and glycolysis. The structural changes in the fecal microflora induced by AFB1 are similar to that observed in IBD (inflammatory bowel disease) patients. The combination of metabolic dysfunction, loss of IECs and glandular atrophy caused by AFB1 can lead to chronic intestinal diseases (Zhou, Tang & Wang, 2021).

The gastrointestinal system in poultry is especially sensitive to AFB1. Broilers fed with different doses of AFB1 exhibit severe damage to the intestinal villi characterized by lower density and absorption area (Yunus et al., 2011; Kana, Teguia & Tchoumboue, 2010), increased atrophy and shedding, and a significant reduction in height (Yin et al., 2016). Moreover, the jejunum of chickens exposed to AFB1 showed histopathological changes, increased apoptosis rates, and altered expression levels of death receptors, endoplasmic reticulum (ER) molecules and apoptotic factors (Zheng et al., 2017). Ingestion of AFB1-contaminated feed can also affect the absorption capacity of the small intestine and impair its innate immune function (Wang et al., 2018c). Furthermore, infiltration of inflammatory cells into the small intestine leads to muco-enteritis (Kumar & Balachandran, 2009). Ducks fed with AFB1-contaminated corns showed longer and wider jejunum villi, which was accompanied by lower average daily gain (ADG) and average daily feed intake (ADFI), resulting in reduced growth and development. Furthermore, the relative weight of the digestive organs, the activity of digestive enzymes and the biochemical indices of intestinal development were also altered (Feng et al., 2017). Thus, AFB1induced intestinal damage can restrict development, disturb the intestinal microflora, and lead to metabolic disorders or chronic intestinal diseases.

Nephrotoxicity

AFB1 is absorbed by the kidneys, and its accumulation in the renal tissues leads to the up-regulation of p21 by MYC, PLK1 and PLD1, resulting in S-phase cell cycle arrest and renal injury (Huang et al., 2019). HEK293 cells treated with AFB1 showed increased apoptosis and DNA fragmentation, which corresponded to the up-regulation of p53, Bax and caspases (Dlamini et al., 2021). AFB1 and AFM1 synergistically increased oxidative stress and the apoptosis pathway in renal cells by regulating the expression level of L-proline (Li et al., 2018). Furthermore, the combination of AFB1 exposure and low protein diet additively reduced weight gain and promoted renal dysfunction in rats, and exacerbated oxidative stress (Rotimi et al., 2018). Exposure to AFB1 and DON synergistically increased oxidative stress in the liver, kidney, and spleen of carp by upregulating Nrf2. Interestingly, ROS generation occurred earlier in the kidneys compared to the liver and spleen (Kövesi et al., 2020). In conclusion, the nephrotoxicity of AFB1 is mainly manifested as oxidative stress induced by p21, L-proline, Nrf2 and other genes. Moreover, AFB1 can have synergistic nephrotoxic effects along with nutrient level and other mycotoxins.

Neurotoxicity

Intragastric administration of AFB1 once weekly for 8 weeks significantly impaired brain function in rats by inducing pathological changes in the cerebral cortex and hippocampus (Bahey, Abd Elaziz & Gadalla, 2015). Long-term exposure to AFB1 may allow it to penetrate the blood–brain barrier, resulting in neurotoxic effects and even chronic neurodegeneration such as that observed in Alzheimer’s disease (Alsayyah et al., 2019). AFB1 inhibits proliferation of human astrocytes by inducing cell cycle arrest and mitochondria-dependent apoptosis (Park et al., 2020). Environmental AFB1 exposure may trigger neuroinflammatory responses by activating the microglia, and increase the susceptibility to neurodegenerative diseases (Mehrzad, Hosseinkhani & Malvandi, 2018a). The neurotoxic effects of AFB1 exposure have also been observed in zebrafish embryos by nuclear magnetic resonance (NMR) (Zuberi et al., 2019). AFB1 exposure decreased the survival rate of embryos by inhibiting oligodendrocyte development (Park et al., 2020). In addition, neuroblastoma cells (IMR-32 cell line) treated with AFB1 also presented intracellular ROS accumulation, DNA damage, S phase arrest and apoptosis (Huang et al., 2020). To summarize, AFB1 can inhibit neural cell development, promote apoptosis, disrupt the homeostasis of the nervous system, and increase the susceptibility to neurodegenerative diseases.

Immunotoxicity

Oxidative stress and apoptosis play key roles in AFB1-induced immunotoxicity (Chen et al., 2013). Oral administration of AFB1 downregulated IFN and TNF in the spleen of mice, increased IL4 levels, and damaged the thymus and spleen, eventually resulting in an impaired immune function (Jebali et al., 2015). AFB1 exposure in rats increased ROS generation and secretion of pro-inflammatory cytokines (TNFα) in the liver cells (Mohammad et al., 2017), which are conducive HCC development (Qin et al., 2016). AFB1 exposure affected transcription of key functional genes in human microglia cell line (CHME5) and human monocyte-derived dendritic cells (MDDCs), and increased apoptosis (Mehrzad, Hosseinkhani & Malvandi, 2018a; Mehrzad et al., 2018b). Similarly, AFB1 treatment decreased the viability of the mouse macrophage RAW264.7 cells in a dose- and time-dependent manner by increasing production of ROS and malondialdehyde (MDA) and decreasing GSH levels. These changes correlated with upregulation of NOS2, TNFα and CXCL2 mRNAs, and downregulation of CD86. AFB1-induced oxidative stress in macrophages also impaired the mitochondrial respiratory chain, leading to activation of the inflammatory response pathways (Ma et al., 2021). AFB1 also decreased the phagocytic capacity of 3D4/21 cells, and induced apoptosis, pro-inflammatory cytokine secretion, DNA damage and oxidative stress. In addition, 3D4/21 cells treated with AFB1 expressed high levels of DNA methyltransferase DNMT1 and DNMT3a, which led to the activation of the JAK2/STAT3 signaling pathway. Inhibition of p-JAK2 and p-STAT3 by blocking DNMT1 and DNMT3a alleviated AFB1-induced immunotoxicity (Zhou et al., 2019). The combination of AFB1 and ochratoxin A (OTA) increased production of TNFα and IL6 in these cells, and decreased lactate dehydrogenase secretion and the phagocytotic index in a concentration-dependent manner. In addition, the combination treatment significantly decreased the production of GSH, increased ROS levels, and promoted IκBa degradation, NF-κB phosphorylation and nuclear translocation of NF-κB. Thus, AFB1 and OTA can synergistically aggravate immunotoxicity by activating of the NF-κB signaling pathway (Hou et al., 2018). AFB1 also impaired the physiological functions of freshly isolated swine alveolar macrophages (SAM), and consumption of AFB1-contaminated feed increased the risk of secondary infections in pigs (Pang, Chiang & Chang, 2020). In addition, AFB1 activated the release of heterophil extracellular traps (HETs) in chicks, and induced the expression of TNFα, IL6 and IL1β, iNOS, COX2, NLRP3, caspase1, caspase3 and caspase11, which resulted in liver and kidney damage (Gao et al., 2022).

Studies show that intake of low doses of AFB1 in animals adversely affect immune organs, decrease antibody titers and complement activity, and cause lymphoid tissue damage. Monogastric animals, especially poultry and pigs, are more sensitive to AFB1-induced immunotoxicity. Broilers exposed to AFB1 showed increased apoptosis of thymocytes due to mitochondrial and death receptor-mediated signaling pathways, as well as DNA damage (Peng et al., 2016). AFB1 also induced tissue damage, cell cycle arrest (Hu et al., 2018) and apoptosis in the Bursa of Fabricius of broilers, which damaged their immune system (Yuan et al., 2016). Ingestion of AFB1-contaminated feed significantly decreased the body weight and lymphocyte activity of pigs, increased the levels of pro-inflammatory cytokines such as TNFα, IL6, IL1β and IFN γ in the spleen, and decreased that of the regulatory factor IL10 (Meissonnier et al., 2008). The immunotoxic effects of AFB1 in livestock may decrease the efficacy of vaccines and increase disease prevalence. These findings indicate that AFB1 affects immune cells and organs, and its immunotoxicity depends on the mitochondrial signaling, death receptor, endoplasmic reticulum, apoptosis, cell cycle and inflammatory pathways.

Reproductive toxicity

AFB1 can impair spermatogenesis through oxidative stress and mitochondria-dependent apoptosis. Mice exposed to AFB1 show impaired blood-testis barrier due to lower levels of the BTB-related junction protein, increased apoptosis in the testes, and the oxidative stress-mediated p38 MAPK signaling pathway, which ultimately affected spermatogenesis (Huang et al., 2021). In addition, AFB1 exposure decreased spermatogenesis in mice by inducing oxidative stress, decreasing mitochondrial content, and upregulating the pro-apoptotic Bax, p53 and Caspase 3 (Yasin, Mazdak & Mino, 2018). AFB1 also induced testicular damage and testicular dysfunction in Dorper rams (Lin et al., 2022), and impaired spermatogenesis in white leghorn cockerels (Ashraf et al., 2022). Exposure to low concentrations of AFB1 for several hours can decrease spermatozoa motility, hyperpolarize mitochondrial membranes, and increase DNA fragmentation (Komsky-Elbaz, Saktsier & Roth, 2018). In addition, AFB1 interferes with porcine oocyte maturation by inducing epigenetic modifications, oxidative stress, excessive autophagy, and apoptosis (Liu et al., 2015a), and affects early embryo development through oxidative stress, DNA damage, apoptosis, and autophagy (Shin et al., 2018). AFB1 can cross the placental barrier in humans (Wild et al., 1991), and AFB1 exposure during pregnancy or lactation can adversely affect the health of the mother as well as the infants. In addition, pregnancy in mice can modulate both phase I and II metabolism, and alter the biological potency of AFB1, thereby increasing liver damage (Sriwattanapong et al., 2017). Prenatal exposure to AFB1 reduces the body weight of neonatal rats, disrupts lipid and hormone levels, and affect the methylation levels of p53 and growth-regulator H19 in the liver and serum. The pathological changes may increase the risk of HCC in the offspring (Rotimi et al., 2021). These studies indicate that exposure of sexually mature animals to AFB1 affects gamete production, gamete quality, and gamete maturation, and AFB1 also disrupts embryonic development, posing a long-term health threat to both pregnant animal mothers and offspring. The reproductive toxicity of AFB1 depends on oxidative stress, DNA damage and repair, apoptosis, autophagy, and epigenetic modification.

Molecular Mechanism of AFB1 Toxicity

AFB1 and multi-omics analysis

Transcriptomic analyses have helped elucidate the mechanisms underlying the pathological effects of AFB1. A total of 1,452 differentially expressed genes (DEGs) have been identified in the liver tissues of AFB1-treated versus healthy mice. Gene Ontology (GO) and Kyoto Encyclopedia of Genes and Genomes (KEGG) pathway enrichment analysis showed that the DEGs were enriched in functions such as cell adhesion, cell proliferation and cell cycle regulation (Xu et al., 2001). In addition, several lncRNAs involved in the regulation of genes related to apoptosis and DNA repair were upregulated following AFB1 exposure (Shi et al., 2016). Transcriptomic analysis of AFB1-exposed rat tissues indicated that the genes affected by AFB1 were mainly enriched in the p53 signaling pathway, bladder cancer-related signaling pathways, inflammatory response, antioxidant response, cell proliferation, and DNA repair. Metabolomic analysis showed that AFB1 dysregulated gluconeogenesis and lipid metabolism (Lu et al., 2013). AFB1 also induced transcriptomic changes in the genes related to cancer development, apoptosis, inflammation, biological activation, and detoxification in the bovine fetal hepatocyte-derived cell line (BFH12) (Pauletto et al., 2020). Finally, the transcriptomes of AFB1-exposed laying hens showed an upregulation of genes involved in hepatic fat deposition and hepatocyte apoptosis, including those related to the mTOR, FoxO, PPAR, fatty acid degradation and fatty acid metabolism pathways (Liu et al., 2020).

AFB1 and the encoding genes

AFB1 can alter the expression levels of oncogenes (such as ras and c-fos) and tumor suppressor genes (such as p53 and survivin) (Su et al., 2004; Duan et al., 2005), and induce genomic instability and mutations by forming DNA adducts, inhibiting DNA repair enzymes and increasing ROS production. It is bio-transformed to AFBO via cytochrome p450 enzymes, which then forms the carcinogenic adducts. In fact, the p53 gene is mutated in the majority of AFB1 induced HCC cases (Engin & Engin, 2019). In a recent study, transcriptomics and functional genomics identified p53 as the critical transcription factor driving the DNA damage response after exposure to benzo (A) pyrene and AFB1 (Smit et al., 2017).

The transcription factor Nrf2 regulates antioxidant/stress response genes and detoxification genes, and Nrf2 knockout rats are highly sensitive to AFB1 (Taguchi et al., 2016). Primary broiler hepatocytes (PBHs) and broiler cardiomyocytes (BCMs) showed a significant decrease in viability, increased mitochondrial dysfunction, ROS generation and high apoptosis rates following AFB1 treatment, all of which are mediated by the Nrf2 pathway (Liu & Wang, 2016; Wang et al., 2017).

Caveolin-1 (CAV1) is a key mediator of AFB1-induced hepatotoxicity. The human hepatocyte L02 cell line showed a marked decline in viability due to increased apoptosis and oxidative stress after AFB1 exposure, which was accompanied by increased expression of CAV1. The latter mediates AFB1-induced oxidative stress through its interaction with Nrf2, leading to the downregulation of cellular antioxidant enzymes and activation of apoptotic pathways. In addition, CAV1 regulates AFB1-induced autophagy via the EGFR/PI3K-AKT/mTOR signaling pathway. Taken together, CAV1 plays a crucial role in AFB1-induced hepatocyte apoptosis by regulating oxidative stress and autophagy, and is therefore a potential therapeutic target against AFB1-related hepatotoxicity (Xu et al., 2020).

AFB1 induces COX2 expression, promotes mitochondrial autophagy and impairs mitochondrial lipid metabolism in hepatocytes, leading to hepatic steatosis (Ren, Han & Meng, 2020). In addition, AFB1 can induce apoptosis and trigger an “eicosanoid and cytokine storm” in the liver, which can initiate tumor growth. Furthermore, AFB1-generated cellular debris can upregulate COX2, soluble epoxide hydrolase (sEH) and ER stress-response genes (BiP, CHOP and PDI) in macrophages (Fishbein et al., 2020). Meanwhile, the expression of COX2 during pregnancy is crucial, and exposure to AFB1 may induce physiological changes controlled by COX2 (Zhu, Tan & Leung, 2016).

AFB1 and non-coding RNAs

Non-coding RNAs play regulatory roles in mediating the toxic effects of AFB1. For instance, miR-429 and miR-24 are up-regulated in liver cancer tissues exposed to AFB1, and correlate to tumor size. Overexpression of both increased AFB1-DNA adducts in the cancer cells, inhibited apoptosis and promoted their proliferation. Thus, miR-429 and miR-24 are reliable biomarkers of AFB1-induced liver cancer (Huang et al., 2013; Liu et al., 2014). Other miRNAs significantly associated with HCC include hsa-miR-96-5p, hsa-miR-30a-3p, hsa-miR-34a-5p, hsa-miR-34b-5p, hsa-miR-222a-3p, hsa-miR-199a-3p and hsa-miR-4286 (Zhang et al., 2019b). Ectopic expression of miR-34a-5p in liver tumor-bearing rats led to cell cycle arrest by inhibiting MET, CCNE2 and CCND1, and promoted repair of AFB1-induced DNA damage in the liver (Liu et al., 2015b). High expression of miR-33a and miR-34a down-regulated the Wnt/β-catenin signaling pathway in AFB1-stimulated HCC cells (Fang et al., 2013; Zhu et al., 2015). In addition, overexpression of lncRNA H19 promoted HCC cell proliferation and infiltration (Lv et al., 2014). MiR-138-1* inhibits the proliferation, colony formation, migration, and invasion of P50 B-2A13 cells (immortalized human bronchial epithelial cells stably expressing CYP2A13), and mediates AFB1-induced malignant transformation by targeting 3-phosphoinositide-dependent protein kinase-1 (PDK1) (Wang et al., 2016b). MiRNA-429, miRNA-24, miRNA-122, miRNA-33a, miRNA-34a-5p, miRNA-300b-3p, miRNA-138-1 and miRNA-34a regulate AFB1-induced tumorigenesis via the GSK-3b–C/EBPa–miR-122–IGF-1R regulatory circuitry, p53 DNA repair axis, and the Wnt/β-catenin signaling pathway (Dai et al., 2017). AFB1-induced fat accumulation and apoptosis in the liver is regulated by the network of lncRNAs, miRNAs and protein-coding genes. Co-expression of lncRNA (TU45776), gga-miR-190a-3p and Bcl-6 gene induce apoptosis in liver cells, whereas co-expression of lncRNA (TU10057), gga-miR-301a-3p, gga-miR-301b-3p and PPARG gene cause fatty liver (Liu et al., 2020). Taken together, AFB1 regulates the expression of coding genes and non-coding RNAs involved in cell proliferation, apoptosis, and carcinogenesis (Fig. 3).

Figure 3 The molecular mechanisms of AFB1 toxicities.

Coding genes, non-coding RNAs and signaling pathways involved in AFB1-induced toxicity. AFB1 induces cell cycle arrest, apoptosis, oxidative stress, ER stress and autophagy through multiple genes, non-coding RNAs and signaling pathways. ROS, reactive oxygen species; ER stress, endoplasmic reticulum stress. Made in ©BioRender (https://biorender.com/).

Conclusions and Future Direction

The contaminated of agricultural produce with AFTs is practically unavoidable worldwide. The presence of AFTs in feeds may decrease feed intake, damage health and affect livestock productivity. In addition, the toxic residues in animal products (milk, meat, eggs) may have some adverse effects on human health. Contamination of plant and animal-derived food products by AFB1 is a major health concern. In this review, we summarized the hepatotoxic, enterotoxic, nephrotoxic, neurotoxic, immune-toxic and gonado-toxic effects of AFB1. In addition, the mechanisms underlying the toxicity of AFB1, including cell cycle arrest, apoptosis, oxidative stress, ER stress and autophagy, and the effector genes, non-coding RNAs and signaling pathways were also discussed.

The toxicity of AFB1 is very complex, and is closely related to the dose, exposure duration, administration mode, solvent, species, gender, age, target organs and so on. More comprehensive and systematic tests are needed to ascertain the acute and chronic AFB1 exposures doses in different species, develop standard dosing methods, and determine the quantified dose–response relationship in vitro and in vivo. In addition, the combined effects of AFB1 with other mycotoxins and with macronutrient deficiency, and well as the underlying mechanisms will be our future research concerns. The relationship between gut microorganisms and sensitivity to AFB1, and the impact of intestinal microbiota on AFB1 metabolism will also be an important research focus. Elucidating the molecular mechanisms underlying the toxicity of AFB1 can help mitigate or eliminate the toxic effects of AFB1 by genetic methods, which will also be worth investigating.

Supplemental Information

Supplemental Information 1 Summarized the number of literatures searched via databases

The number list of articles searched via databases.

Click here for additional data file.

Additional Information and Declarations

Competing Interests

Author Contributions

Data Availability

The authors declare there are no competing interests.

Congcong Li conceived and designed the experiments, performed the experiments, analyzed the data, prepared figures and/or tables, authored or reviewed drafts of the article, and approved the final draft.

Xiangdong Liu conceived and designed the experiments, prepared figures and/or tables, authored or reviewed drafts of the article, and approved the final draft.

Jiao Wu performed the experiments, authored or reviewed drafts of the article, and approved the final draft.

Xiangbo Ji performed the experiments, authored or reviewed drafts of the article, and approved the final draft.

Qiuliang Xu performed the experiments, authored or reviewed drafts of the article, and approved the final draft.

The following information was supplied regarding data availability:

Raw data are available in the Supplementary Files.

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
