# Peer review of "Research progress in toxicological effects and mechanism of aflatoxin B1 toxin"

_PeerJ, doi:10.7717/peerj.13850_

## Round 0.1 · original submission · Major Revisions

Major revisions are required.

Reviewer 1 ·

Basic reporting

No comment

Experimental design

No comment

Validity of the findings

No comment

Additional comments

This review article is well written. I suggest some minor changes to improve it.
It is better to include paragraph about measures to reduce contamination of food with mycotoxins. Also, paragraph about the proper care of domestic animals after exposure to mycotoxins.
In figures; there should be figure legend at which all abbreviations used in the figure are written in full term.

Reviewer 2 ·

Basic reporting

This review article suffers from a number of major flaws in some of the areas covered that calls into question the accuracy with which other aspects of the pathobiology of aflatoxin are appropriately summarized from the literature. There are over 1300 review articles on aflatoxin and its toxicology and this particular manuscript does provide a broader perspective on this compounds impact on a number of organ systems other than liver. However, there are examples of completely incorrect statements from the literature such as on lines 118 and 119 that purports that some of the aflatoxin metabolites can be converted in the sulfates. This is absolutely incorrect with respect to aflatoxin M1 and with respect to aflatoxin Q1 and aflatoxin P1 these are excreted as glucuronides. Further, the citation of the article online 699 with respect to p53 mutations is not appropriate since this is not the first report of these aflatoxin specific mutations that were initially published in Nature in 1991. These types of errors reflect poorly upon the summarization of other data in the submitted manuscript.

Experimental design

As stated above, the final selection of the nearly 120 papers summarized in this review article appears to be biased and not illustrating a curating process to surface the best examples of a given area. Since there are so many reviews available these authors might wish to focus on the most current literature over the past 10 years rather than trying to summarize the nearly 6 decades of work on aflatoxin.

Validity of the findings

The several examples already stated that are fundamentally incorrect in this review article calls into question the care and accuracy with which other literature is summarized.

Reviewer 3 ·

Basic reporting

The document aims fit the scope of the journal and the topic is still of interest. To my knowledge there is no review that compiles all the data that authors tried to summarize in the present paper. However, I believe that although the paper is interesting, lot of work must be done before publishing it. The paper needs in general a revision of English and technical vocabulary, particularly in the introduction and discussion. Introduction and conclusion are poorly developped.

Lines 56-61: writing should be improved by using technical language such as organoleptic properties.

Lines 86-87: I hardly recommend rephrasing this phase, authors are doing this review based on studies.
Lines 107-148: this part is clear; I suggest specifying in which organisms this pathway occurs.
Line 330: redundant to say male cockerels. The authors may just keep taking about male chickens to avoid confusion.
Line 346: pregnant mothers is commonly use for human beings only
Lines 432-435: should be checked.

Experimental design

The current phrasing makes comprehension difficult, some affirmations are not completely true, some of the references should be changed, the parts in which results from different studies are shown should be double checked to be sure of proper paraphrasing (some examples provided below).

Line 47: AFT are produced by several species, the most important and overspread are those cited in the article, yet they are not the only ones.
Line 49: species that produce AFT are more frequent in the tropics and subtropics, so this phrase must be rephrased as it is not true.
Lines 62-64: Mold produce 4 AFT (AFB1, AFB2, AFG1, AFG2), so these are not derivates.
Line 45: Report of 25% proper citation FAO 1999
Line 68: the reference of AFB1 as powerful carcinogenic belongs to the reports from IARC, EFSA, JECFA since the 1999, so I strongly suggest using one of them as the proper reference, specially IARC and EFSA
Line 82: a citation is missing
Line 711: Modifier Hyglene par Hygiene

Figure and tables:Figure should be worked, as present now they are not clear and the aim of figures are not so clear. The lack of legends makes difficult to understand the aim of the figures.

Figures:
Figure 1: To my knowledge AFM1, AFP1 and AFQ1 are not completely excreted after been transformed. AFM1 is known to have some toxic effects in mammals and in some to be bioaccumulated.
Figure 2: it is not clear to me the aim of the authors with it, nor why chickens are presented twice.
Figure 3: I do not get why it is shown a maize corn and the fungi on the top, why both?
Tables:
The tables are quite messy. I suggest to organized them, taking into account the information of the columns. For the species column it should be organized for species, at the moment is a mix of species, gender and age (example for chickens, they are presented as chicks, broiler, cockerels). The doses can also be shown as intervals to make to table more useful. For me it is not clear the solvent column, Does it refer to the extraction solvent?

Validity of the findings

Conclusion is weak and do not show all the work performed through the text.
The arguments are not developped and do not set the expectations.

Reviewer 4 ·

Basic reporting

Overall the information presented represents valuable information regarding toxic effects of AFB1 in humans and animals along with its toxicokinetic properties. The review is well written, treats an actual problem and concerned on it with details (complexity of AFB1 toxicity) and provides reliable references for future studies. But, I have some comments please address it.

Experimental design

no comment

Validity of the findings

no comment

Additional comments

1. Line 33: gives should be giving
2. Line 43: Productions not produce
3. Line 47: Aspergillus flavus, A. parasiticus, should be italic
4. Line 50: 80-90% not 80%-90%
5. Line66: the citation ( there is a missing reference in the entire text)
6. Line 81: add and before the last reference
7. Line 82: citation (a missing reference)
8. This reference is missing inside the text
Gartung A, Wang Y, Bielenberg DR, Huang S, Kieran MW, Hammock BD, and Panigrahy D. 507 2020. Resolution of eicosanoid/cytokine storm prevents carcinogen and inflammation-initiated 508 hepatocellular cancer progression. Proc Natl Acad Sci U S A 117:21576-21587. 509 10.1073/pnas.2007412117
9. The references are not arranged alphabetically (line 506 and 510 and so on…..)
10. All figures missed caption, it is better to write the full name of the abbreviations under the figures to be more descriptive and understood without having to refer to the main text.
11. In table (1): in immune-toxicity, 50% ethanol is not used in vivo for mice as a solvent; mice are more sensitive to ethanol-induced lethality. Please revise it again.

---

## Round 0.2 · accepted · Accept

The review has been accepted for publication.

Reviewer 2 ·

Basic reporting

The revised manuscript has addressed all of the major critiques in my initial review.

Experimental design

All prior concerns have been adequately addressed.

Validity of the findings

The latest manuscript has updated references and been more inclusive of prior literature.

Additional comments

There are still several minor typographical errors, e.g. lines 397 and 661, that a final editing should correct.

Reviewer 4 ·

Basic reporting

The authors edited all the comments and responded to all the comments in details so I recommend accepting the manuscript

Experimental design

No comment

Validity of the findings

No comment

Additional comments

No